# Using Community Based Research Frameworks to Develop and Implement a Church-Based Program to Prevent Diabetes and Its Complications for Samoan Communities in South Western Sydney

**DOI:** 10.3390/ijerph18179385

**Published:** 2021-09-06

**Authors:** Dorothy W. Ndwiga, Kate A. McBride, David Simmons, Ronda Thompson, Jennifer Reath, Penelope Abbott, Olataga Alofivae-Doorbinia, Paniani Patu, Annalise T. Vaovasa, Freya MacMillan

**Affiliations:** 1School of Health Sciences, Western Sydney University, Penrith 2751, Australia; 18029879@student.westernsydney.edu.au; 2Institute of Health and Management, Parramatta 2150, Australia; 3School of Medicine, Western Sydney University, Penrith 2751, Australia; k.mcbride@westernsydney.edu.au (K.A.M.); da.simmons@westernsydney.edu.au (D.S.); r.thompson@westernsydney.edu.au (R.T.); j.reath@westernsydney.edu.au (J.R.); p.abbott@westernsydney.edu.au (P.A.); 4Diabetes Obesity Metabolism Translational Research Unit, Western Sydney University, Penrith 2751, Australia; 5Translational Health Research Institute, Western Sydney University, Penrith 2751, Australia; 6Powell Street Family Practice, Yagoona 2199, Australia; droad@powellstmedical.com; 7The Practice Blacktown NSW, Blacktown 2148, Australia; paniani.patu@bigpond.com

**Keywords:** Australian Samoan community, community-based participatory research, diabetes, obesity, health promotion, community activation

## Abstract

Pasifika communities bear a disproportionate burden of diabetes compared to the general Australian population. Community-based participatory research (CBPR), which involves working in partnership with researchers and communities to address local health needs, has gained prominence as a model of working with underserved communities. This paper describes how Le Taeao Afua (LTA) Samoan diabetes prevention program was underpinned by two CBPR frameworks to develop a culturally tailored church-based lifestyle intervention to prevent diabetes and its complications in the Australian Samoan community. The name LTA, which means ‘a new dawn,’ was chosen by the community to signify a new dawn without diabetes in the Australian Samoan community. Strategies for engaging with the Australian Samoan community in South Western Sydney are discussed mapped to the key principles from the CBPR frameworks. In particular, this paper highlights the steps involved in building relationships with Samoan community leaders and the vital role of community activators and peer support facilitators in the success of delivering the program. Lessons learnt, such as the importance of church and maintaining a Samoan way of life in daily activities, and processes to build effective partnerships and maintain long-term relationships with the Australian Samoan community, are also discussed. Our paper, through providing a case example of how to apply CBPR frameworks, will help guide future community-based health promotion programs for underserved communities.

## 1. Introduction

There are ~335,000 Pasifika people in Australia [1], with New South Wales (NSW) having the largest proportion of people of Pasifika ancestry of all the states and territories [1,2]. Samoans comprise the second largest Pasifika community (~75,000) (after New Zealand Māori) in Australia [1], with many having migrated from New Zealand. In NSW, the majority of Samoans reside in South Western Sydney (SWS) [1,2]. Similar to other culturally and linguistically diverse (CALD) communities in Australia, Samoans experience disproportionately higher rates of chronic conditions compared to their counterparts of European descent [3,4]. Studies conducted in Samoa [5,6], Australia [7,8,9], New Zealand [10] and the USA [11,12], have found that Samoans experience poor health outcomes and high prevalence of health conditions, including diabetes and obesity. Type 2 diabetes has been identified as one of the leading causes of death among American Samoans in the USA [13] and of disability in the Independent State of Samoa [14]. In Australia, Samoans living in Queensland have been found to be seven-times more likely to be hospitalised for diabetes complications than the general population [9]. There is strong evidence that while Samoans have a genetic predisposition to obesity and diabetes [15,16] their increased morbidity is also associated with lifestyle related factors [8].

Despite these existing health disparities, in Australia and beyond, Samoans are under-represented in health research. Few studies have focussed on Samoans living in Australia, highlighting a gap in supporting a rapidly increasing population identified to be at increased risk of obesity and diabetes [7,8,17]. Community empowerment and participatory approaches to research have been recommended to help address the health inequities that exist in CALD communities [18,19,20,21,22]. Community-based participatory research (CBPR) frameworks can be used to guide research focused on addressing health inequities by supporting ways for local communities to work collaboratively with researchers as ‘equal partners’ in identifying and tackling prioritized health needs [18,22]. CBPR has been used successfully among both Indigenous and CALD communities in Australia and internationally [23,24,25,26].

### Community-Based Participatory Research (CBPR)

CBPR, as well as associated methodologies such as action research, participatory research and participatory action research, has gained prominence as an approach for conducting research in underserved communities [18,22]. Establishing partnerships with underserved communities leverages cultural expertise to enhance community participation in research that is meaningful to the community, and assists in addressing health disparities [23,26]. For example, a community-based study among Marshallese community members with diabetes in the USA, which involved a community-research coalition, demonstrated a significant reduction in HbA1c [24]. This study was a diabetes self-management family-based education intervention aimed at improving glycaemic control and was delivered by Marshallese community health workers (CHWs) and diabetes nurse-educators. From other research among the same Marshallese community [27], the Marshallese CHWs were involved in the design of the program, but no information was provided on whether the nurse-educators were involved in the design of the project or only the delivery aspect of the project. Community-based participatory research is not just about delivery but also initiation, development of the research topic and community capacity building, where researchers work through community partnerships ‘with’ rather than ‘on’ a community [28]. This is viewed by some as a contradiction of ‘traditional’ research [22]. Traditional approaches to research usually involve researchers identifying a health issue, formulating research questions, and determining how interventions will be delivered, evaluated and results disseminated. 

Instead, CBPR involves a partnership between researchers, organisations and community members where all partners are accountable in all phases of the research process [28]. Figure 1 highlights the nine key principles of CBPR. The CBPR approach recognises the unique contributions of each stakeholder and allows shared decision-making, where the researchers and community members work together in identifying a local health problem of concern and solving the health problem together [29]. This can include formulating the research questions, developing the study design, implementing the study design, analysing and interpreting the data, disseminating the results and making recommendations for future research, policy and practice [28]. This partnership empowers and creates a sense of ownership in local communities and therefore supports long-term sustainability of the program. In CBPR, the relationship between the researchers and community members is developed on trust and mutual understanding over time [22]. Although, the core principles of CBPR partnerships may be similar across academic-community alliances, each community setting will have its own strengths and challenges unique to the social context and characteristics of that community [30]. 

Following an initial approach by the Sydney Samoan community to our research team, we utilised a CBPR approach to create a successful partnership to co-design and co-lead a program of research aimed at preventing diabetes and its complications in their community. We also aimed to build community capacity and address Samoan community health needs while developing an ongoing, sustainable partnership of mutual trust and respect with the wider Australian Pasifika community. This partnership between the South Western Sydney (SWS) Australian Samoan community and researchers at Western Sydney University (Western) began in October 2015. The partnership led to the creation and delivery of a culturally tailored evidenced based diabetes prevention pilot program, which consisted of delivery of 12 healthy lifestyle messages and 10 diabetes management messages for those diagnosed with diabetes. The pilot study aimed to determine the effectiveness of a diabetes prevention program in the Samoan community with the goal of conducting a larger step-wedged RCT with all Pasifika communities in Sydney. 

The program’s 12 healthy lifestyle messages were designed to influence healthy lifestyle choices and were originally co-developed with New Zealand Māori [31] based upon earlier work evaluating previous programs with New Zealand Māori, Samoan and Tongan communities [10,32]. Additionally, the 10 diabetes messages aimed to improve management of diabetes and reduce risk of diabetes complications. Papers reporting on the baseline characteristics of participants [7] and outcomes of a lifestyle intervention pilot study [33] from the resulting Le Taeao Afua (LTA) Samoan diabetes prevention program pilot project have already been published. To develop a culturally appropriate diabetes prevention program, a study focusing specifically on the experiences of Samoan people living with type 2 diabetes and perceptions of diabetes, including how cultural beliefs and life experiences affected diabetes outcomes was conducted [34]. The project and the research leading up to it were guided by the nine principles of CBPR as described by Israel et al., 2013 [18] (Figure 1) and learnings from studies conducted on Indigenous communities in New Zealand [26] (Figure 2). Table 1 illustrates the CBPR approaches we utilised in the research partnership process with the Australian Samoan community. 

This article aims to describe the CBPR approach which guided the development of LTA Samoan diabetes prevention program. We provide our experiences and characterise the process of conducting CBPR with the Australian Samoan community to assist others in future who are working collaboratively with Samoan or other underserved communities on health promotion programs. 

## 2. Materials and Methods

Ethical approval was obtained from Western prior to project commencement (H11699 and H11388). Qualitative data were collected in three forms: (1) via one-to-one interviews with community members/leaders before the intervention, (2) interviews with participants post intervention and (3) minutes from reference group meeting discussions. A summary of participant data collection processes and numbers of participants is included in Figure 3.

### 2.1. Formative Interviews

Diabetes was identified as a health concern among the Australian Samoan community living in SWS following meetings with a Samoan general practitioner (GP) and researchers at Western. Semi-structured qualitative interviews were subsequently conducted with Samoan community members prior to designing LTA program to develop a community-driven research agenda and to better understand Samoan community members’ perceptions of diabetes and their lifestyle behaviours. The interviews with the community leaders also aimed at seeking suggestions for strategies to target diabetes and its complications in the community. These participants were recruited through one general practice in SWS. The interviews were conducted by a Pasifika (Samoan and Tongan background) co-author (AV) from the university. 

### 2.2. Implementation Process Data

A Samoan community reference group was initiated to guide and oversee the research program. Samoan representatives in the reference group ranged from high school students through to community elders, and also included the Samoan Consul General and the Samoan High Commissioner. These individuals represented a diverse range of multidisciplinary employment areas including church pastors, social work, media and general practice. Minutes from reference group meetings between November 2015 and November 2019 were used in this analysis. Although the community leaders in the formative stage were recruited from one general practice, the Samoan community reference group that followed had attendees that spanned a wider network and were not all just from this one general practice. Reference group participants were any Samoans that heard of the project through community leaders about our project who volunteered their time to guide the work.

### 2.3. Evaluative Interviews

To gather participant perspectives on LTA program, semi-structured interviews were conducted on completion of the first 3–8 months of the program. Interviews explored participants’ knowledge about diabetes and program effectiveness, including participant experiences and changes to diet and physical activity made as a result of participating in LTA program. Interviews were conducted in English by three Western research students (not of Samoan background). A bilingual researcher (fluent in Samoan and English) was available to provide clarification of any Samoan idiomatic expressions as required. 

### 2.4. Data Analysis

All interviews were conducted in English, audio recorded and transcribed verbatim, with integrity checks offered to participants to check for accuracy. Some reference group meeting components were discussed in Samoan, but all minutes were translated and written in English. Quirkos qualitative software was used for data management [35] to thematically analyse interview transcripts and meeting minute data. A framework analysis method [36] informed the analysis of the data with main patterns (themes) emerging from the data aligned with the nine stages of the CBPR framework [18] and recommendations by Voyle and Simmons [26]. The results below include a discussion of how the two CBPR frameworks were used to inform LTA diabetes prevention pilot project. ‘P’ after each quote represents individual participant and their participant number.

## 3. Results

We conducted 15 semi-structured interviews with Samoan community leaders prior to designing LTA program and 10 interviews on completion of the first 3–8 months of the program. Eleven community reference group meetings were held during the course of LTA program. 

**Principle** **1.***CBPR recognises community as a unit of identity*.

The Samoan culture revolves around the concept of *fa’aSamoa,* which refers to the traditional Samoan way of life or Samoan way of doing things [37]. The *fa’aSamoa* is both a hierarchical and collective culture with roles and responsibilities influenced by social status, gender and age [19,38]. In respect to this, the reference group recognised key community facilitators—community leaders who were influential members of the community, such as church pastors. However, other community representatives, including high school and university students, were also invited to be part of LTA and contribute to decisions throughout the intervention process. Including young people was viewed as important, particularly because they can influence their parents to make healthier choices: 

*“I think to help our young people because they have more understanding, they have the language now, they are more into the, the normal community now, …but I’m sure helps with the attitude of parents at home”*. P112

The partnership between the Australian Samoan community and researchers at Western recognised the community’s cultural identity and identified the church as an appropriate setting for the program. This was through the guidance of the reference group and qualitative interviews from the Samoan leaders and other Samoan community members. Undertaking these formative interviews was essential for engaging with the Samoan community to understand how to best undertake the project, and to strengthen the CBPR partnership by ensuring that LTA was tailored to the needs of the local Samoan community. The partnership was also aimed at enhancing community ownership of the research, and to support the implementation and maintenance of the lifestyle intervention through seeking and following community advice. Several community members suggested that the church was the best venue for a diabetes prevention program as church pastors were seen as people of authority held in high esteem:

*“…. if it was down to like pastors and ministers of churches because I think they play a huge role especially when you have a congregation of 200 people and whatever the minister says everyone does hey! So it would be good to have them on board coz they have huge influence”*. P107

**Principle** **2.***CBPR builds on strength and resources within the community*.

In order to address the health concern (diabetes) it was important to first identify the strengths and resources within the community. The reference group recognised the strengths and assets of the community (e.g., established churches, individual skills of volunteers within churches), which were useful resources for the LTA program. Four churches were recommended by the reference group to run LTA pilot study. These churches were selected on the basis of having a large congregation (one church) and the Seventh Day Adventist (SDA) denomination as it had an established health temperance committee (three churches). It was intended after discussions with the reference group that any church programs delivered by the temperance committee would run simultaneously with the LTA church program. Church volunteers (Peer Support Facilitators [PSFs]) of Samoan culture were also invaluable resources in implementing LTA. The use of community volunteers to build capacity among Samoans as part of the pilot project was both recommended and appreciated by community members:

*“…. because our people, especially the old age, they can’t understand because you know, because there are some other medical words they cannot understanding but we if we are having, bringing in those people, those people, they can talk in the language, might be the message of delivering, they’re going to get the message”*. P111

*“The use of PSFs was a great idea and great value. These are people from a cultural perspective and leaders are held in high esteem”*. P117

The use of community volunteers to deliver the intervention was also an ideal approach as volunteers had in-depth knowledge of community issues. They were also well positioned to assist their peers to lead healthier lives due to their understanding of contextual factors affecting the community uptake of a healthy lifestyle. For example, a Samoan community member identified the need of having leaders to guide the community to engage in healthy lifestyles:

*“We need good leaders because I think often times Samoans are good followers…. so if we are able to find good leaders that are able to lead our people to good dieting and you know, introducing good food and be able to lead them by way of healthy cooking and things like that I think we can overcome”*. P104

Furthermore, diabetes was recognised as an extensive problem, therefore PSFs who have contextual understanding of the health issue could relate to the common experiences of their peers and provide invaluable social and emotional support. Samoan community members spoke of there being strong motivation to intervene to protect families within the community, particularly children:

*“I look at it as most of the, each individual families, kind of having one person they are affected with diabetes. You can see most of the children they are getting obesity and they are having diabetes at a young age. It’s very sad. It’s very sad”*. P111

**Principle** **3.***CBPR facilitates a collaborative, equitable partnership in all phases of research, involving an empowering and power-sharing process that attends to social inequalities. Gaining access and building trust with the community*.

For the project to be successful, researchers at Western had to gain and build trust with the Samoan community within the Greater Western Sydney area, and particularly in the SWS area where the research commenced. One Samoan GP (a community ‘champion’) initiated the development of a relationship between the Samoan community and the academic research team, to address her concerns about high rates of diabetes in her community. She arranged for a meeting with key Samoan community representatives and researchers at Western to discuss potential collaboration on diabetes related projects. As the Samoan GP was well connected within the Samoan community, she approached several key Samoan community representatives. From this initial meeting, a Samoan community representative reference group was established to guide diabetes research undertaken in partnership with the University. No constraints were set on who could join this reference group, allowing a diverse range of community perspectives to be included. Co-chairs from the community were identified (one male, one female) to facilitate discussions. Invitation to reference group meetings came via our community champion and the co-chairs, which meant that the community ultimately had control of who was invited to attend. The reference group consisted of regular representatives, as well as ‘floating’ representatives who attended only some meetings, hence meetings often had new attendees and were attended by 25–30 attendees at a given time. The inclusion of floating and regular attendees was both a benefit and a challenge for updating the community on project development. New representation often brought different viewpoints and suggestions. However, a large part of the meeting was often focused on reviewing what had already been carried out and why, rather than on progress to date and ways to move forward. 

Meetings were held at a location decided on by the Samoan reference group, at times this included university premises as a reflection of the shared partnership. Emphasis on collaboration between Samoan community members and researchers at Western as equal partners and having a community oversight of the research were crucial in developing and sustaining the partnership. The reference group members were involved in planning of LTA program including provision of guidance and support around: identification of churches to be invited to participate in the pilot study; development of effective appropriate recruitment strategies; ensuring ethical approaches in recruitment and intervention delivery from the perspectives of the community; and identifying two Samoans for employment to deliver the intervention. The reference group approved the data collection tools [31,32], once the validated tools had been laid out in a format that was visually appealing (e.g., incorporation of pictures throughout): 

*“…. visuals are very good, you’ve got to make it interesting you know, all that. I love the chart…... I understood all that”*. P102

Resources such as data collection tools that were to be included in the program implementation were found to be acceptable and participants reported that the resources encouraged them to take on the program healthy lifestyle messages:

*“For us, it’s a very good program and even our food from now on with my family, ….my wife just buy a packet of salad, green salad, just put it in the fridge and the bread, make a sandwich and cheese, that’s our lunch, with my family and my sons. This worked like that. We changed the red meat, more salad, sugar. We’re not drinking anymore fizzy drink….”* P119

LTA program resources were also translated into Samoan as suggested by the reference group. These translated resources ensured effective delivery of the lifestyle messages by the PSFs:

*“When I volunteered, I was worried about language barrier but the messages from our coaches (community activators) were easy to understand. I could deliver the messages of the intervention in Samoa”*. P121

To ensure the fidelity of the intervention was not compromised by members starting their own community-based intervention, the specific content of the intervention was not detailed at the reference group meetings. Prior to commencement of LTA program, a Memorandum of Understanding (MoU) between the University and each of the participating churches was signed. The MoU was intended to ensure that all parties were aware of their role and responsibilities and to further develop trust between the local Samoan community and reinforce the commitment of ongoing relationships of the university with the local Samoan community. 

**Principle** **4.***CBPR fosters co-learning and capacity building among all partners. CBPR integrates and achieves a balance between knowledge generation and intervention for the mutual benefit of all partners*.

**Principle** **5.***Designing the LTA intervention—integrating the Samoan perspectives*.

To ensure LTA diabetes program was culturally tailored, LTA program incorporated Samoan values and knowledge with involved partners recognising that both parties brought diverse skills, expertise and different perspectives and experience to the partnership process [18]. The Samoan community provided cultural viewpoints and insight into the Samoan way of life, whilst the researchers brought research experience, obtained funding for the project from various funding partners and provided program resources and training to the Samoan community. The reference group was also taught about the research processes including seeking funding to undertake the project. For the research to be undertaken and be successful multiple partners were required including partners who funded the entire research project. Therefore, co-funding was sought from several local health districts (LHDs) in Greater Western Sydney (GWS) following the pilot, further demonstrating commitment through different partnerships to improve the health and well-being of the Samoan and wider Pasifika communities. 

A community activation approach underpinned LTA with an emphasis on collaborative partnership between the researchers at Western and the Samoan community. Interviews highlighted the need to build capabilities within community members to support healthy lifestyles:

*“…. it’s better to go to the people and let our people facilitate those, those who can share the information and deliver in a way that they feel that we are all part of it. We’re not there to teach them, just to go there and raise awareness and the effect of so many people die young because of diabetes”*. P112

Community activation strategies involved recruitment of community members through a call for expressions of interest for church congregants to volunteer to deliver the lifestyle program, supported by two Samoan community activators (CAs). This process enhanced community capacity building by mobilising the community to implement and sustain the lifestyle program. The LTA diabetes program provided healthy alternatives and/or modifications of traditional recipes. Interview participants stressed the need for culturally appropriate Samoan-tailored resources:

*“Perhaps even having like posters of our own people, you know, like pushing it out that way, and then just maybe having like a number for our people to call you know, for information that’s available to them”*. P101

To achieve this in the pilot study, the CAs and one community member guided adaptation to intervention materials and translation of key documents. The volunteer peer support (PSFs) who delivered the program within churches received 1–2 days of workshop training (some PSFs had the 2-day workshop condensed into one training day due to their time restraints) delivered by the CAs. The topics covered in the workshop included: understanding of the study protocol; upskilling PSFs in delivering peer support, confidentiality; an introduction to intervention messages; and responsibilities and support available to PSFs. All PSFs who completed the training received a certificate. The certificates were awarded by the Samoan Consul general at a community reference group meeting. This community capacity concept was vital to acknowledge and ‘celebrate’ the PSFs who had volunteered to be leaders in their community. The training provided valuable skills to PSFs that ensured that the Samoan community was empowered, owned the diabetes project and had the skills and knowledge to continue with the program once the study ended. To further strengthen the skills of the PSFs and build ongoing community capacity building, the CAs also provided ongoing support to the PSFs as well as led group sessions when requested by churches and offered one to one coaching to participants that were struggling to meet their lifestyle goals. 

**Principle** **6.***CBPR emphasises local relevance of public health problems and on ecological perspectives that attend the multiple determinants of health*.

The CBPR process addressed the public health concern (diabetes) brought forward by the community champion (Samoan GP), and what was highlighted as an important health issue by interview participants and the reference group. CAs and PSFs helped the community members to understand the magnitude of diabetes and empowered the community to make healthier choices. The reference group members helped researchers to understand the Samoan culture which was fundamental in having a successful partnership and diabetes prevention program:

*“The University has also benefit from the contribution of the Samoan Community. We have shared our Samoan culture, beliefs, traditions and understandings—and you have ensured culturally appropriate processes and developed an intervention that has been language clear and culturally acceptable. While maintaining the importance of a healthy lifestyle change. This has been well received by our community”*. P121

The reference group and LTA participants discussed both the barriers and facilitators that contribute to the diabetes outcomes in the Samoan community. Issues identified though consultation with the reference group meetings and interviews with community members prior to commencement of the LTA program included denial/embarrassment about having diabetes, Samoan culture such as the Sunday feast, lack of culturally tailored resources, and work commitments resulting in many Samoan not seeking appropriate health care:

*“The other thing is of course, for Samoans, it’s very difficult to share what problems you have. You can easily say I’m a diabetic, but they won’t tell you whether they are suffering from the usual associated you know, problems with diabetes, and so by holding in that information, people are not getting the right diagnosis”*. P105

*“I think the fact that people are less active and our food, and it’s rude to say no. So, and proportions are big, I think doesn’t help……even though they try and bring exercise programs into the church and encourage, it’s still, it’s very revolved around food and you can see the bottles of soft drinks on the table…”*. P103

Further, LTA participants shared some similar experiences reported in wider lifestyle health promotion literature, such as lack of time, lack of motivation and scheduling of LTA data collection times, as barriers to participating in the diabetes program: 

*“Perhaps make data collection to be more efficient as it dips into schedules of participants. Data collection was performed on activity nights, which was to the annoyance of multiple church members who only wished to enjoy the activities”*. P122

One aim of the pilot was to refer participants to their GP. During data collection in LTA, participants with clinical measurements of blood pressure, random blood glucose levels and HbA1c that were out of normal range were given a referral letter to take to their GP. This was intended to ensure LTA participants received urgent treatment. One participant described how he went to see their GP after receiving a referral letter from LTA after data collection session:

*“The intervention is good. I was given a letter by LTA because my sugar was high. It was the first time I see a doctor in a long time. I went to see my doctor and said I have diabetes and confirmed I have diabetes”*. P124

**Principle** **7.***CBPR involves systems development using a cyclical and iterative process*.

This principle of CBPR involves putting the knowledge gained in the research partnership into action drawing on the strengths and experiences of each partner. In our research, the findings from LTA were used to develop a funding proposal for a larger study to include all Pasifika communities. The reference group was involved in all processes of this new research including presenting their experiences of LTA to potential funders and having Pasifika representation on the funding investigator team. Additionally, the Samoan reference group members in LTA offered solutions to barriers encountered by the research team while conducting the intervention which have informed the larger study. For example, recruitment of participants was slow initially in LTA and community feedback through our reference group and direct from church members highlighted that the written recruitment materials were not effective on their own—they asked for more visual materials and face to face interactions:

*“Samoan people are not reading people. They traditionally pass information down orally and visually. We should look at having simplified (resources) and have someone speak in a video…”*. P117

In response, videos were created with the CAs describing in English what was involved in the research and why it was being carried out, with Samoan sub-titles included on screen. Additionally, researchers and the community activators attended church ceremonies to present on the research and answer questions from the community. These strategies boosted recruitment considerably. To assist with empowering the community, during the recruitment phase reference group meetings were held for the community to develop a project logo and title. A vote was held to select the logo that best suited the aspirations of the research and that represented the Samoan community. 

**Principle** **8.***CBPR disseminates results to all partners and involves them in the wider dissemination of results*.

Findings of the intervention were disseminated through the reference group, which all community members participating in the program were invited to attend. The reference group also involved community presentations, dancing and food to highlight the benefit of LTA to their community. Individual data were also available to any participant who requested this. In line with the CBPR approach Samoan community members were involved with research outputs and were invited and included as co-authors of publications and co-presenters at meetings and conferences. Further, participants discussed their experiences and the impact LTA had on their community with local funding partners. This process facilitated the links between the community and the external funding organisations.

**Principle** **9.***CBPR involves a long-term process and commitment to sustainability*.

The LTA intervention framework was designed to ensure that the intervention was sustainable once the research ended. Through building community capacity by training PSFs, a pool of skilled individuals able to provide peer support within churches now exists. The larger step-wedged randomised control trial (RCT) following LTA is evaluating the same intervention framework but with CAs embedded in current Local Health District health promotion/population health units to provide ongoing support for the volunteers within churches. LTA pilot churches sit within the areas covered by the wider project and will therefore benefit from the LHD embedded CAs ongoing. Following completion of LTA pilot study, the churches adopted lifestyle messages into their church program. For example, all churches established weekly walking groups and adapted a healthier menu in their Sabbath feast, showing sustainability of the intervention within the pilot churches themselves: 

*“We have made changes to the Sabbath community meal. We used to enjoy large meal and have plenty of food. But with LTA we have learnt new techniques and started eating lighter meals. We have agreed to this and now eat outside to enjoy each other. PSFs are leading our own Zumba classes, weight friendly classes”*. P121

## 4. Discussion

The CBPR frameworks [18,26] recommend strategies and principles for addressing health disparities in underserved communities and we successfully applied these frameworks to establish a strong coalition between researchers at Western and the Australian Samoan community in SWS. This paper described the CBPR and partnership process to set a community-driven research agenda. It provided a contextualisation of the key CBPR principles involved in the design and implementation of a diabetes prevention program in SWS delivered by Samoan CAs and volunteer PSFs. 

The aim of the Western-Samoan community partnership was to develop a long-term, sustainable and beneficial relationship founded on mutual trust and respect to improve the health and well-being of Samoans living in SWS. Community consultation to understand the needs of the community and involvement in community health project planning demonstrates respect to the community members [39,40]. Based on previous research [23,26] and the lessons learnt from our research, the sustainability of health programs require partnerships founded on respect for the community in focus. In LTA, engaging with Samoan community leaders and community members throughout the research process from planning through to implementation and dissemination of the program findings, demonstrated respect through acknowledging the needs of the community. Most Pasifika people including Samoans identify as Christians [2], and one of the major components of *fa’aSamoa* is a strong belief in God [38]. On up-rooting to other countries, churches help maintain the *fa’aSamoa* [41]. The church affiliation is an important aspect of *f**a’aSamoa* and on migration with the absence of the traditional village, the Samoan churches are regarded as ‘urban villages’ that provide specific cultural activities and networks [38,42]. Following consultations and recommendations from the Samoan community LTA was delivered as a church-based lifestyle intervention with the intervention integrated within the church activities. This ensured LTA diabetes program captured a large Samoan audience and was culturally appropriate. Additionally, church-based approaches have been recommended and successfully used to promote healthy lifestyles among Samoans in New Zealand and the USA [10,17,37]. 

A trusting partnership should be established from the very beginning by including the community representatives in the planning aspects of the health programs. This creates a sense of ownership of health programs if the community is involved in determining and addressing their main health concerns. Engagement with the community to determine how to deal with the health concern should ensure that the program proposed is culturally sensitive and meets the needs of the target community. To ensure that LTA diabetes program was culturally tailored to the health needs of the Samoan community, we conducted qualitative interviews and consulted with community members to understand their interests and adapt our existing diabetes program as it was developed and implemented. As was carried out prior to LTA, to ensure community health preventative health programs are successful, future research should utilise formative evaluations with community members. Formative evaluations have the greatest influence when they begin in the early stages of program development [43] and are recommended to enhance program sustainability and ensure resources are allocated efficiently [26]. Additionally, to increase sustainability of the program, researchers should consider use of peer support and actively engage with local health care professionals and community leaders from target populations in the design and implementation of health promotion programs. Use of peers would involve training the peers to deliver the program and this would increase community capacity and empower the community to run similar programs in future. In our diabetes research program, a call for expressions of interest to church congregations to volunteer and deliver the lifestyle intervention was made. The volunteer PSFs received 1–2 days of workshop training delivered by community activators and covered the following topics: understanding of the study protocol, introduction to intervention lifestyle messages, confidentiality and upskilling in delivering peer support. Church volunteers (Peer Support Facilitators [PSFs]) of Samoan culture were also invaluable resources in implementing LTA. Use of PSFs in LTA built community capacity and ensured the intervention was delivered in a culturally relevant and sustainable way. Use of PSFs ensured the program was sustainable as community members received training to continue with the project once the project funding had stopped. Use of volunteer peers to influence healthy behaviours has been recommended and successfully used among several CALD communities [7,17,32,44,45]. 

Alongside this qualitative work discussed in this article, data on the effectiveness of the community-based diabetes project were collected throughout the pilot study. In the pilot study, data were collected using questionnaires and clinical measurements. Clinical measurements included body mass index (BMI) derived from height and weight measurements, HbA1c, body fat, random blood glucose (RBG), heart rate and exercise capacity (assessed using the six-minute walking test). The pilot study showed that the community-based diabetes lifestyle interventions resulted in beneficial changes in diabetes knowledge and clinical outcomes, such as HbA1c and systolic blood pressure. Findings from the pilot study have been published elsewhere [7,33]. During the research process and in our partnership with the Samoan community, we identified and addressed key considerations in the pilot project including the role of hierarchy, spirituality, and use of culturally tailored resources. These key components are now being further tested in an ongoing large step-wedged randomised controlled trial among all major Pasifika communities in Sydney. The diabetes prevention program in the step-wedge trial is being conducted in 48 Sydney Pasifika churches. Future researchers should seek to expand collaborations with Pasifika communities and integrate community-driven research agendas. These collaborations/partnerships should focus on and reflect the needs of communities. Engaging with communities demonstrates respect and willingness to learn from the community and can lead to developing and implementing community-driven sustainable interventions to improve their health.

### Strengths and Limitations

This is the first study that describes how the CBPR framework was utilised to conduct a community-based lifestyle intervention to prevent diabetes and its complications in the Australian Samoan community. However, there were limitations. Participants were not contacted to review their transcripts, which may have helped further validate findings. However, two of the authors (RT and OA-D), both Samoan, acknowledged and confirmed interpretation of study findings as being representative of their experience of community beliefs. Also, those participating had an interest in the program and hence may not have been representative of community leaders as a whole. Additionally, the participants were recruited from one area in Sydney (SWS) and may not be a true representation of the entire Samoan community. 

## 5. Conclusions

Overall, the Australian Samoan community in SWS was committed to the Western and Samoan community collaboration research process aimed at tackling diabetes and its complications in their community. The implications for individuals as well as the Samoan community were significant as the community strived to maintain healthy lifestyles. The Samoan community showed enthusiasm for the broader LTA vision of an era without diabetes in the Australian Samoan community. Working in partnership facilitated the development of a culturally acceptable program that addressed a local health need in the Samoan community living in SWS. Given the health disparities that still exist and the desire of the Samoan community to improve their health, we hope to continue our work with the Samoan community (and other Pasifika communities) to enable all Pasifika communities in Australia to gain greater control over their own health.

## Figures and Tables

**Figure 1 ijerph-18-09385-f001:**
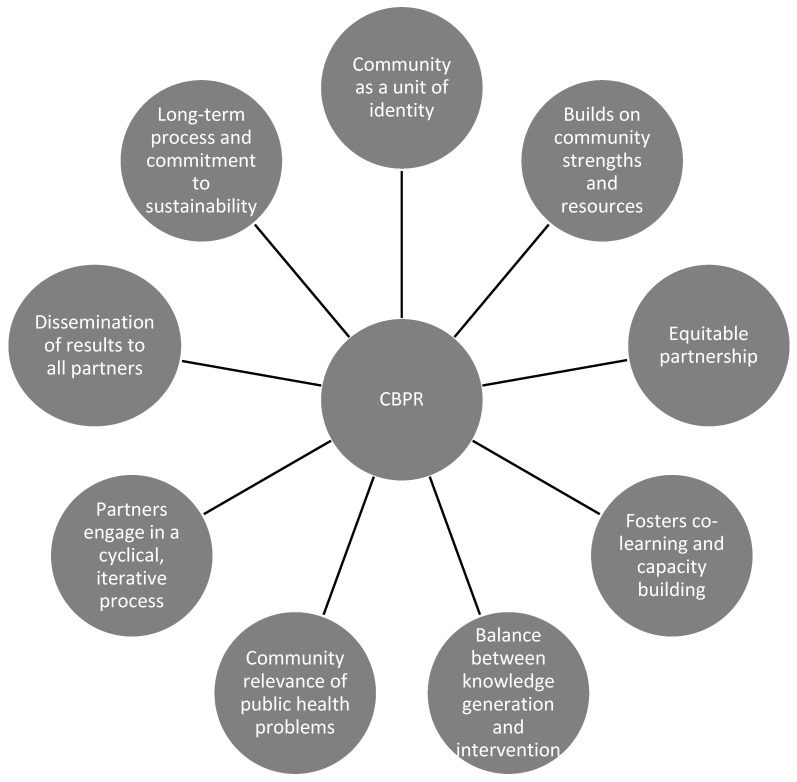
The key principles of Community–Based Participatory Research (CBPR). Adapted from “Methods for Community-Based Participatory Research for health” [18] (pp. 9–11).

**Figure 2 ijerph-18-09385-f002:**
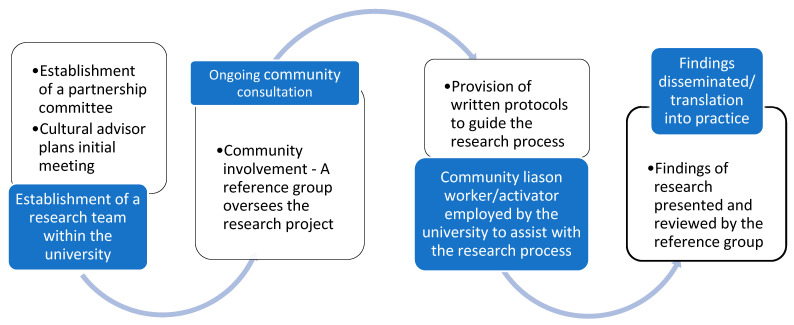
Key steps for formative and process evaluation of a community development partnership. Adapted from Voyle & Simmons, 1999.

**Figure 3 ijerph-18-09385-f003:**
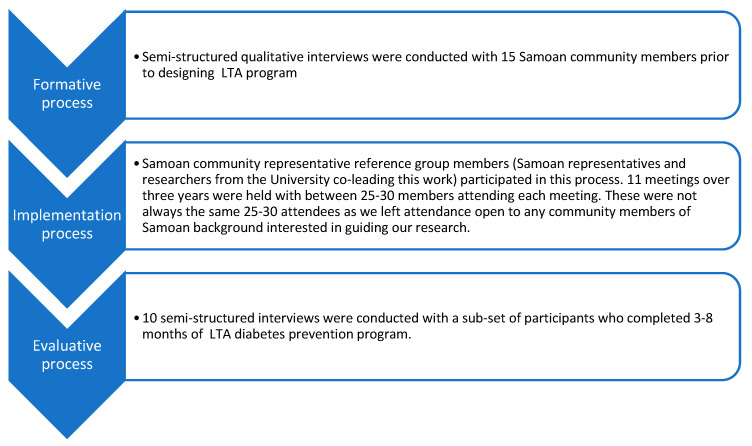
Data collection process.

**Table 1 ijerph-18-09385-t001:** Community Based Participatory Research (CBPR) principles utilised in the community engagement process with the Australian Samoan community.

CBPR Principles by Israel et al., 2013 Applied in the Research Partnership	CBPR Approaches Utilized from Previous Learnings from Henderson et al., 2002; Voyle & Simmons, 1999.
*1. CBPR acknowledges community as a unit of identity.* *Le Taeao Afua* recognised and included key members of community such as church pastors and community elders, as equal partners throughout the research process.	*1. Establishment of a partnership committee to identify health priorities and develop specific projects with the help of a cultural advisor. The cultural advisor plans initial meetings with community members and University researchers.* A Samoan GP aware of previous work conducted by researchers at the University with Samoan communities in New Zealand, approached these researchers due to her concerns of the high prevalence of diabetes in her community. The Samoan GP, a leader within the Samoan community in Sydney, arranged for an initial meeting between key community representatives and the University researchers. A community reference group was initiated following this meeting.
*2. CBPR builds on strengths and resources within the community.* *Le Taeao Afua* identified community assets and resources and built on the strengths within the community, such as established local churches and individual skills (PSFs).	*2. Establishment of a research team within the University to carry out tasks set by the partnership committee.* To ensure the recommendations from the community members/community reference group were implemented, a research team comprising of researchers from the University and Samoan community activators (once recruited into the project) was initiated.
*3. CBPR facilitates a collaborative, equitable partnership in all phases of research, involving an empowering and power-sharing process that attends to social inequalities*. A partnership between the Samoan community and university partners was developed. To ensure a community-engaged research a Samoan community reference group was initiated to oversee the research process. The reference group consisted of Samoan leaders/community members and researchers from the university.	*3. Provision of written protocols (Memorandum of Understanding (MoU)) to guide the research process. The MoU should be developed in consultation with the community*. The MoU is intended to guide both parties in the research process and develop trust for ongoing relationships by ensuring that roles and responsibilities are defined. A MoU was developed in collaboration with the Samoan community reference group and the University and was signed by each participating church and the University before the start of the intervention.
*4. CBPR fosters co-learning and capacity building among all partners.* *Le Taeao Afua* acknowledged that both partners (Samoan community and researchers) had diverse skills and expertise. Community members were trained to deliver the intervention promoting community capacity within the Samoan community and researchers received research training where necessary and updated their cultural competence skills.	*4. Community consultation—The community must be consulted about any research project being proposed to be conducted within it. Approval to undertake the project must be sought from community gate keepers. A representative from the local community should be nominated to monitor the implementation of the MoU.* Permission and approval to undertake the diabetes research program was sought from the key Samoan community leaders during and following the initial meeting organised by the Samoan GP. *The community should be notified of the financial costs of conducting the research process* Community representatives attending the reference group were notified of funding applications to undertake the project and the costs of developing, delivering and evaluating the program. In-kind contributions were also described so that the community were aware of what the research team and research partner organizations were providing to support the program.
*5. CBPR integrates and achieves a balance between knowledge generation and intervention for the mutual benefit of all partners.* *Le Taeoa Afua* gathered information from community members prior to conducting the intervention and incorporated the community knowledge in planning and delivery of the intervention. *Le Taeoa Afua* built community capacity by training and involving volunteer peer supporters to deliver the intervention. The program also ensured a balance between community benefits with addressing the research aim and objectives. For example, participants were required to notify the CAs if they were involved in other lifestyle interventions rather than excluding them from the research.	*5. Community involvement—A reference group should be initiated to oversee the research project. Roles of the reference group should be made with respect to specific cultural and social needs. Regular reference group meetings should be held to ensure the committee is up to date with the research process, obstacles, issues and to give feedback.* A Samoan community reference group comprising of 25–30 members was initiated following a meeting with key community representatives. Meetings were held every 3–5 months. The reference group co-chairs decided who to send invitations to from within their communities. New members were welcomed through invitation from the reference group members.
*6. CBPR emphasises local relevance of public health problems and on ecological perspectives that attend the multiple determinants of health.* *Le Taeoa Afua* addressed a local health problem identified by the Samoan community. To ensure that the research was successful, *Le Taeao Afua* discussed both barriers and facilitators to a healthy lifestyle in the community, with the aim of developing strategies to overcome these barriers during the research process and promote better outcomes by empowering community.	*6. A community liaison worker/activator should be employed by the university to assist with consultation, data collection and analysis. The reference committee should be involved in the recruitment process of a cultural person who represents the community at the university. For community and individual empowerment peer volunteers should be trained and utilized in the research process.* The community reference group oversaw the recruitment and employment of two bilingual Samoan community activators (fluent in English and Samoan) by the University to deliver the intervention.
*7. CBPR involves systems development using a cyclical and iterative process.* Findings from *Le Taeoa Afua* were used to develop a larger study with the aim of reducing the impact of diabetes and its complications in the wider Pasifika communities in Sydney.	*7. Data storage and retention—The research project should be conducted in accordance with data principles and agreed protocols with the community.* *Individual privacy and confidentiality should be maintained, and the community should not have access to individual data.* Research data was stored on password protected university computers and databases. Individual data were available to participants on their request. Individual data was not shared with anyone else. Publication of research data and reports from *Le Taeao Afua* were approved by designated community representatives. All publications arising from *Le Taeao Afua* included details of the joint research process between the Samoan community and research team at the university. Research findings were disseminated at community reference group meetings.
*8. CBPR disseminates results to all partners and involves them in the wider dissemination of results.* Findings of *Le Taeao Afua* were disseminated to all members of the community reference group and community members. Individual data were available on request to all participants and overall study findings were available to reference group members to distribute to their respective audiences.	
*9. CBPR involves a long-term process and commitment to sustainability.* *Le Taeao Afua* was designed to ensure that the intervention was sustainable and able to address multiple determinants of health. Following the initial pilot study focusing on diabetes, other health initiatives have been introduced to meet the needs of the community.	

## Data Availability

The data are not publicly available due to confidentiality protection, and ethical obligations of ensuring the anonymity of participants involved in qualitative research.

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
