# Peer review of "Using Community Based Research Frameworks to Develop and Implement a Church-Based Program to Prevent Diabetes and Its Complications for Samoan Communities in South Western Sydney"

_ijerph, 2021, doi:10.3390/ijerph18179385_

Round 1

Reviewer 1 Report

The Authors of the article have raised a very important issue worthy of attention.

Below are some comments and suggestions:

  1. The methodology is convolutedly written. In my opinion, it would be beneficial to include a diagram showing the different stages along with the number of participants at each stage and the duration of each stage.
  2. I think you should include tables of the results in the results section.
  3. The discussion is written too generally, it should be expanded. Additionally, the discussion would benefit from pointing out the strengths and weaknesses of the research conducted.
  4. The pages are incorrectly numbered.

Author Response

Response to Reviewers comments – IJERP

Reviewer 1

Associate Editor Comments

Response

1. The methodology is convolutedly written. In my opinion, it would be beneficial to include a diagram showing the different stages along with the number of participants at each stage and the duration of each stage.

Thank you for your comment and we greatly appreciate your time reviewing our manuscript.

As requested, we have included a flow diagram illustrating the data collection process and the number of participants in each phase. More information on participants is included in the results section.

The following table has been added to the manuscript:

Line 217-218, page 6: “Flow diagram 1. Data collection process

 “

2. I think you should include tables of the results in the results section.

Thank you for your comments, we have a table that provides details of each of the stages of the CBPR framework. Additionally, we have our results section written to reflect the stages of the CBPR framework and adding another table would be a repetition of the current summary of the stages of CBPR utilised in our research. Table 1 on pages 14-17 illustrates the CBPR approaches utilised in our research partnership process with the Australian Samoan community and our results sections have been written to reflect the CBPR framework by Israel et al., and  Voyle & Simmons.

No change made at this point-happy for the editor to advise

3. The discussion is written too generally, it should be expanded. Additionally, the discussion would benefit from pointing out the strengths and weaknesses of the research conducted.

Thank you for your comment. We have expanded our discussion section by adding more information and included a section outlining the strengths and weaknesses of our study.

The following has been added to the manuscript:

Line 518-532, page 12:

Based on previous research [23,26] and the lessons learnt from our research, the sustainability of health programs require partnerships founded on respect for the community in focus. “In LTA, engaging with Samoan community leaders and community members throughout the research process from planning through to  implementation of the program and dissemination of the program findings, demonstrated respect through acknowledging the needs of the community. Most Pasifika people including Samoans identify as Christians [2], and one of the major components of fa’aSamoa is a strong belief in God [38]. On up-rooting to other countries, churches help maintain the fa’aSamoa [41]. The church affiliation is an important aspect of fa’aSamoa and on migration, with the absence of the traditional village, the Samoan churches are regarded as ‘urban villages’ that provide specific cultural activities and networks [38, 42]. Following consultations and recommendations from the Samoan community LTA was delivered as a church-based lifestyle intervention with the intervention integrated within the church activities. This ensured LTA diabetes program captured a large Samoan audience and was culturally-appropriate. Additionally, church-based approaches have been recommended and successfully used to promote healthy lifestyles among Samoans in New Zealand and the USA [10, 17, 37].”

Lines 552-564, page 12:

Additionally, to increase sustainability of the program, researchers should consider use of peer support and actively engage with local health care professionals and community leaders from target populations in the design and implementation of health promotion programs. Use of peers would involve training the peers to deliver the program and this would increase community capacity and empower the community to run similar programs in future. “In our diabetes research program a call for expressions of interest to church congregations to volunteer and deliver the lifestyle intervention was made. The volunteer PSFs received 1-2-days of workshop training delivered by community activators and covered the following topics: understanding of the study protocol, introduction to intervention lifestyle messgaes, confidentiality, upskilling in delivering peer support. Church volunteers (Peer Support Facilitators [PSFs]) of Samoan culture were also invaluable resources in implementing LTA. Use of PSFs in LTA built community capacity and ensured the intervention was delivered in a culturally relevant and sustainable way. Use of PSFs ensured the program was sustainable as community members received training to continue with the project once the project funding had stopped.” Use of volunteer peers to influence healthy behaviours has been recommended and successfully used among several CALD communities [7,17,32,44,45].

Lines 565-584; Lines 584-585, page 13:

“Alongside this qualitative work discussed in this article, data on the effectiveness of the community-based diabetes project were collected throughout the pilot study. In the pilot study, data were collected using questionnaires and clinical measurements. Clinical measurements included body mass index (BMI) derived from height and weight measurements, HbA1c, body fat, random blood glucose (RBG), heart rate and exercise capacity (assessed using the six-minute walking test). The pilot study showed that the community-based diabetes lifestyle interventions resulted in increase in diabetes knowledge and reductions of clinical outcomes such as HbA1c and systolic blood pressure. Findings from the pilot study have been published elsewhere [7, 33].” During the research process and in our partnership with the Samoan community, we identified and addressed key considerations in the pilot project including the role of hierarchy, spirituality, and use of culturally tailored resources. These key components are now being further tested in an ongoing large step-wedged randomised controlled trial among all major Pasifika communities in Sydney. The diabetes prevention program in the stepwedge trial is being conducted in 48 Sydney Pasifika churches. “

Line 591-601, page 13: “Strengths and limitations

This is the first study that describes how the CBPR framework was utilised to conduct a community-based lifestyle intervention to prevent diabetes and its complications in the Australian Samoan community. However, there werelimitations. Participants were not contacted to review their transcripts, which may have helped further validate findings. However, two of the authors (RT and OA-D), both Samoan, acknowledged and confirmed interpretation of study findings were representative of their experience of community beliefs. Those participating had an interest in the program and hence may not have been representative of community leaders as a whole. Additionally, the participants were recruited from one area in Sydney and may not be a true representation of the entire Samoan community.”

4. The pages are incorrectly numbered

We apologise for that. We have edited the manuscript to reflect the correct page numbers.

Reviewer 2 Report

Title:  Using Community Based Research Frameworks to develop and implement a church-based program to prevent diabetes and its complications for Samoan communities in South Western Sydney

Authors: Dorothy W. Ndwiga, Kate A. McBride, David Simmons, Ronda Thompson, Jennifer Reath, Penelope Abbott, Olataga Alofivae-Doorbinia, Paniani Patu, Annalise T. Vaovasa, Freya MacMillan.

General comment:

Lifestyle interventions, including diet and physical activity, are fundamental in treating diabetes and often determine the effectiveness of pharmacological therapy. Considering that diabetes affects individuals worldwide, it is necessary to individualize the approach and recommendations for patients, taking into account their cultural background. In their work, Dorothy W. Ndwiga et al. describes how the Samoan diabetes prevention program based on the Community Based Research Frameworks helped develop a targeted lifestyle intervention to prevent diabetes and its complications in a high-risk population - the Australian Samoan community. The project aligns with the current trends regarding the individualization of lifestyle interventions based on mutual trust in patients with diabetes. The manuscript is well designed and written; therefore, I have only a few minor comments that should be clarified before the article is accepted for publication.

Minor  revisions:

1) Did the authors of the study plan to verify the effectiveness of the proposed program employing clinical and biochemical tests? It would be advisable to assess whether the implemented actions translated into real health benefits.

2) Material and Methods:

“Semi-structured qualitative interviews were subsequently conducted with Samoan community members prior to designing LTA program to develop a community-driven research agenda and to better understand Samoan community members’ perceptions of diabetes and their lifestyle behaviours. The interviews with the community leaders also aimed at seeking suggestions for strategies to target diabetes and its complications in the community. These participants were recruited through one general practice in SWS.”  – If the community leaders were recruited through one general practice in SWS – were they representative for the whole community?

3) Figure 1. “ Knowledge generation” – is a capital “K” in the word knowledge justified?

Author Response

Response to Reviewers comments – IJERP

Reviewer 2

Associate Editor Comments

Response

1. General comment: Lifestyle interventions, including diet and physical activity, are fundamental in treating diabetes and often determine the effectiveness of pharmacological therapy. Considering that diabetes affects individuals worldwide, it is necessary to individualize the approach and recommendations for patients, taking into account their cultural background. In their work, Dorothy W. Ndwiga et al. describes how the Samoan diabetes prevention program based on the Community Based Research Frameworks helped develop a targeted lifestyle intervention to prevent diabetes and its complications in a high-risk population - the Australian Samoan community. The project aligns with the current trends regarding the individualization of lifestyle interventions based on mutual trust in patients with diabetes. The manuscript is well designed and written; therefore, I have only a few minor comments that should be clarified before the article is accepted for publication.

We thank the reviewer for their positive comments acknowledging the importance of our research project, quality of our manuscript, acknowledging that the study was well designed and paper well written.

2. Did the authors of the study plan to verify the effectiveness of the proposed program employing clinical and biochemical tests? It would be advisable to assess whether the implemented actions translated into real health benefits.

Thank you for your comment. Yes, we have conducted a pilot study among the Australian Samoan community to assess the effectiveness of a community-based diabetes program. We have provided more information in our manuscript that illustrates the success and effectivieness of the CBPR framework utilised to engage witht the Samoan community to develop a community-based diabetes prevention program. Please find the manuscripts that evaluated the effectiveness of Le Taeao Afua diabetes project below [7,33].

Ndwiga, D.W.; McBride, K.A.; Simmons, D.; Macmillan, F. Diabetes, its risk factors and readiness to change lifestyle behaviours among Australian Samoans living in Sydney: Baseline data for church. Health Promot J Austr. 2019, 31, 268-278; doi:10.1002/hpja.276

Ndwiga, D.W.; Macmillan, F.; McBride, K.A.; Thompson, R.; Reath, J.; Alofivae-Doorbinia, O.; Abbott, P.; McCafferty, C.; Aghajani, M.; Rush., E.; Simmons, D. Outcomes of a church-based lifestyle intervention among Australian Samoan in Sydney – Le Taeao Afua diabetes prevention program. Diabetes Res. Clin. Pract. 2020, 160, 1- 11; https://doi.org/10.1016/j.diabres.2020.108000

The following has been added to the manuscript:

Lines 565-579, page 12-13:

Alongside this qualitative work discussed in this article, data on the effectiveness of the community-based diabetes project were collected throughout the pilot study. In the pilot study, data were collected using questionnaires and clinical measurements. Clinical measurements included body mass index (BMI) derived from height and weight measurements, HbA1c, body fat, random blood glucose (RBG), heart rate and exercise capacity (assessed using the six-minute walking test). The pilot study showed that the community-based diabetes lifestyle interventions resulted in increase in diabetes knowledge and reductions of clinical outcomes such as HbA1c and systolic blood pressure. Findings from the pilot study have been published elsewhere [7, 33].

Following the success of the pilot study, a larger randomised trial involving 48 Sydney Pacific churches is currently underway.

The following has been added to the manuscript:

Lines 584-585 page 13:

“The diabetes prevention program in the stepwedge trial is being conducted in 48 Sydney Pasifika churches.”

3. Material and Methods:

“Semi-structured qualitative interviews were subsequently conducted with Samoan community members prior to designing LTA program to develop a community-driven research agenda and to better understand Samoan community members’ perceptions of diabetes and their lifestyle behaviours. The interviews with the community leaders also aimed at seeking suggestions for strategies to target diabetes and its complications in the community. These participants were recruited through one general practice in SWS.”  – If the community leaders were recruited through one general practice in SWS – were they representative for the whole community?

Thank you for your comment. Although the community leaders were initially recruited  from one general practice, the reference group had attendes who were not from the practice. Reference group participants were any Samoans that heard of the project through community leaders about our project that wanted to join in. This made the research project representative. We have updated the satatements in our manuscript to emphasis this point.

The following statements has been added to our manuscript:

Line 176-180, page 5:

However, although the community leaders in the formative stage were recruited from one general practice, the Samoan community reference group that followed had attendees who were not from the practice. Reference group participants were any Samoans that heard of the project through community leaders about our project and wanted to join in.

Line 591-601, page 13: “Strengths and limitations

This is the first study that describes how the CBPR framework was utilised to conduct a community-based lifestyle intervention to prevent diabetes and its complications in the Australian Samoan community. However, there were limitations. Participants were not contacted to review their transcripts, which may have helped further validate findings. However, two of the authors (RT and OA-D), both Samoan, acknowledged and confirmed interpretation of study findings were representative of their experience of community beliefs. Those participating had an interest in the program and hence may not have been representative of community leaders as a whole. Additionally, the participants were recruited from one area in Sydney (SWS) and may not be a true representation of the entire Samoan community.”

3. Figure 1. “ Knowledge generation” – is a capital “K” in the word knowledge justified?

Thank you for bringing this to our attention. The ‘K’ should be lower case. We have edited the manuscript to reflect the change.

Round 2

Reviewer 1 Report

Thank you very much to the authors of this article for all your answers. The article has been refined and I recommend it for publication.